# Adipose Tissue Development Relies on Coordinated Extracellular Matrix Remodeling, Angiogenesis, and Adipogenesis

**DOI:** 10.3390/biomedicines10092227

**Published:** 2022-09-08

**Authors:** Elizabeth K. Johnston, Rosalyn D. Abbott

**Affiliations:** Department of Biomedical Engineering, Carnegie Mellon University, Pittsburgh, PA 15213, USA

**Keywords:** white adipose tissue, brown adipose tissue, angiogenesis, adipogenesis, extracellular matrix remodeling, adipose development, mechanics, paracrine signaling, proteolysis, cell shape

## Abstract

Despite developing prenatally, the adipose tissue is unique in its ability to undergo drastic growth even after reaching its mature size. This development and subsequent maintenance rely on the proper coordination between the vascular niche and the adipose compartment. In this review, the process of adipose tissue development is broken down to explain (1) the ultrastructural matrix remodeling that is undertaken during simultaneous adipogenesis and angiogenesis, (2) the paracrine crosstalk involved during adipose development, (3) the mechanical regulators involved in adipose growth, and (4) the proteolytic and paracrine oversight for matrix remodeling during adipose development. It is crucial to gain a better understanding of the complex relationships that exist between adipose tissue and the vasculature during tissue development to provide insights into the pathological tissue expansion of obesity and to develop improved soft-tissue reconstruction techniques.

## 1. Introduction

While classically viewed as merely a storage depot, adipose tissue (fat) has many roles and is classified as a connective, thermogenic, and endocrine organ. Healthy adipose tissue is required to regulate systemic energy homeostasis, modulate inflammatory responses, store and metabolize steroids, protect internal organs, and maintain body temperature (both as an insulating organ and through non-shivering thermogenesis) [1,2,3,4,5,6]. Proper adipose tissue development relies on tightly regulated processes that require careful coordination and cooperation between many different cell types and their matrix cues. 

Full-term human infants are composed of 11 to 28% fat [7], which is classified as either brown or white fat. Brown fat, which fully develops and matures prenatally, plays a pertinent role in neonatal thermal regulation, as it contains ample mitochondria to convert the many, small lipids stored in the brown multilocular adipocytes into thermal energy [8]. However, brown adipose tissue proceeds from being 5% of the human body mass at birth to a mere 1.5% of the adult human mass and is replaced by white adipose tissue as the primary adipose mass [9,10]. On the other hand, white adipose tissue fluctuates in mass throughout both prepubescent and adult life. White adipose tissue serves as the primary energy reservoir in the human body, with approximately 90% being classified as subcutaneous (underneath the skin) and 10% being visceral (surrounding the internal organs/viscera), although there have been quantitative differences shown to be dependent on sex, menopausal state, and disease state [11,12,13]. 

White adipose tissue is a heterogenous mixture of cells, with mature adipocytes being the primary metabolic component and accounting for approximately 90% of the tissue’s volume [14,15]. These mature adipocytes arise from precursor preadipocytes that exist alongside many other cells, including adipose stem cells, stromal cells, endothelial cells, and resident immune cells. Despite taking up only 10% of the tissue’s space, these cells, termed stromal vascular cells, play an important role in regulating vascular innervation and the immune response within the tissue [16]. The proper development of white adipose tissue relies on highly coordinated spatial and temporal communication amongst all of these cells and their microenvironment. There are several reviews published that focus on the interplay between adipose and endothelial cells during obesity-driven adipogenesis and angiogenesis [17,18]. However, the process of adipose organogenesis relies not only on cellular crosstalk but also on cellular self-assembly, which is highly dependent upon both extracellular and intracellular forces in order to control the cell’s fate, matrix remodeling and mechanics, and the development of tissue patterns [19]. There is accumulating evidence indicating the importance of mechanics given that it affects fundamental factors of proliferation, migration, and differentiation [20,21,22,23]. With the bulk modulus of adipose tissues being one of the lowest in the body, around 0.5–1 kPa [24], proper cell fate relies on controlled proliferation and differentiation amidst softer materials. Furthermore, with adipose tissue being physiologically exposed to a range of bulk physiological forces (compressive, tensile, and shear) due to body weight [25,26], there must be adaptive modalities on a cellular level to maintain bulk tissue health and integrity. 

Here, the process of adipose tissue development is broken down to explain (1) the ultrastructural matrix remodeling that is undertaken during simultaneous adipogenesis and angiogenesis, (2) the paracrine crosstalk involved during adipose development, (3) the mechanical regulators involved in adipose growth, and (4) the proteolytic and paracrine oversight for matrix remodeling during adipose development. 

## 2. Adipose Tissue Remodeling during Fetal Development

### 2.1. Ultrastructural Changes during Adipose Tissue Development

Early studies piloted by Hausman *et al.* and Poissonet *et al.* described the complexities of fetal subcutaneous white adipose tissue development using early microscopical and immunohistochemical techniques [27,28,29,30,31,32,33,34,35]. Collectively, these studies describe fetal adipose tissue growth through five defined morphometric stages, as shown in Figure 1. Notably, these developmental stages are highly dependent upon primitive matrix deposition and vessel outgrowth and organization. 

In studying human buccal (cheek) fat, Poissonet *et al.* found that prior to week 14 of gestation, the fat pad consisted solely of amorphous fibers forming a loose connective tissue [27,28]. The second stage was then distinguished by the existence of adipose lobules with primordial vasculature. This vasculature formed a glomerular structure around mesenchymal cellular aggregates. The third stage was defined between weeks 14 and 16 of gestation, whereby capillaries began to develop within a more mature lobular architecture and mesenchymal cells began to compact, allowing for their early differentiation [27,28]. Small multilocular adipocytes began to appear in stage 4 of the development process, between 23 and 29 weeks of gestation [27,28].

It was shown that during this adipogenic time frame, there was unique basal lamina and glycoprotein development along the microvasculature and around the adipocytes [35]. Notably, in both stage 4 and stage 5 of adipose development, logarithmic adipocyte hypertrophy was occurring alongside expansive, tortuous vascular growth [35]. Hausman *et al.* showed that in this developing adipose tissue, the larger, tortuous capillaries had an abundance of chylomicrons (large triglyceride-rich lipoproteins) and very low-density lipoproteins along their lengths, which were used as substrates for lipoprotein lipase (LPL) [31]. Additionally, these vascular cells had high organelle and intracellular vesicle content, indicating the ability to potentiate transfers requiring more energy, ultimately fueling the early expansion of the tissue. This was further indicated by the expression of Adenosine Triphosphatase (ATPase) along the capillaries [31,32]. It was further found that this tortuosity greatly assisted in rapid adipocyte growth by minimizing the spatial distance between the capillaries and the developing adipocytes [30,35]. In the final stage of adipose development, definitive lobules were surrounded by mesenchyme and fibrous collagen, which rapidly condenses and thickened to form the interlobular septa [27,28,29,30,31,32,33,34,35]. This array of histological and cytochemical studies show that there are immense ultrastructural changes throughout the simultaneous development of adipocytes and the maturation of pre-existing capillary beds [29,30,31,32,33,35,36,37].

### 2.2. Remodeling during Angiogenesis and Adipogenesis 

#### 2.2.1. Angiogenesis

In embryonic development, the main vasculature of the human body is formed through *de novo* vasculogenesis, whereby stem cells undergo differentiation towards endothelial progenitor cells, also known as angioblasts, and then towards endothelial cells to form tubule structures [38]. However, in specified organogenesis, vessels branch off of existing vasculature through controlled sprouting or non-sprouting angiogenesis (Figure 2), with the assistance of existing mesenchymal cells [39]. As previously discussed, the primitive vasculature in adipose tissue has developed prior to week 14 [27,28]. In order for angiogenesis to occur, matrix metalloproteinases (MMP) must first break down the basement membrane that is currently supporting the vessel. In these early stages, MMP2, MMP3, and MMP9 tend to start this process allowing for the release of fibrinogen and fibronectin, which lay the provisional foundation for the migration of additional endothelial cells [40,41,42,43,44]. Chemotactic signals, such as vascular endothelial growth factor (VEGF), encourage the proliferation and migration of existing endothelial cells towards the prepared void [45,46,47]. Furthermore, platelet-derived growth factor-β (PDGF-β) recruits pericytes/smooth muscle cells to stabilize the vessel [45]. From then on, transforming growth factor beta (TGF-β) stimulates the differentiation of mesenchymal stem cells to fibroblasts and mural cells in order to increase extracellular matrix (ECM) production to promote vessel maturation [40,48,49]. While the general composition of the vessel’s basement membrane is rather consistent: Laminin and Type IV Collagen linked with Nidogen and Perlecans [35,50,51], the isoform profile and quantity of each ECM will be unique to the specification of the vessel, whether it be arterial, venule, or capillary. For example, Laminin α4 exists on all vessels while Laminin α5 is more exclusive to capillaries [39,40]. Furthermore, Laminin 10 is present at birth, but gradually, it is replaced with Laminin 8 [39], showing that not only does each cell type have a unique matrix signature but also that this signature changes over time.

#### 2.2.2. Adipogenesis

Adipogenesis is a highly regulated, dynamic process whereby adipose stem cells undergo stepwise differentiation towards an adipocyte. One notable change in white adipocyte development includes the gradual morphological change from the fibroblastic shape to a multilocular adipocyte and then finally a spherical unilocular adipocyte, designed to store triglycerides efficiently and effectively [52]. Throughout this adipogenic process, the cells experience alterations in their ECM secretions and network organization, which ultimately affect the rigidity of the environment [53,54]. Antras *et al.* found that during the differentiation of adipocytes, there was an obvious slowdown and halt in the synthesis of actin and other cytoskeletal proteins to accommodate the expansion and accumulation of lipid droplets intracellularly [55].

One notable change in the ECM profile of adipose-derived stem cells (ASCs) cultured in growth media versus differentiation media is the gradual loss of fibronectin and increase in the secretion of laminin of ASCs undergoing adipogenesis [56]. Several studies have shown that the utilization of decellularized matrix from differentiated adipocytes increased the adipogenic potential of ASCs when compared to the adipogenic potential of stem cells cultured with the decellularized matrix of ASCs grown in growth media, indicating that the matrix contains adipogenic cues [57,58]. Considering their pre-established commitment to the adipocyte lineage, 3T3-L1 cells are a common model to study adipogenesis. In their undifferentiated state, 3T3-L1 cells have a fibroblast-like morphology and secretome [59]. It has been shown that these fibroblasts primarily secrete fibrillar Type I and III Collagen when exogenously stimulated with ascorbic acid [60,61]. In addition to their usage as an adipogenic model, this ability to secrete fibrous matrix has previously made them very beneficial as feeder cells to more fragile cells [62,63]. In these adipogenic models, there is an early peak of fibrillar proteins (Types I, II, and III collagen) by day 4 of differentiation, followed by a transient decline that is offset by a gradual increase in basement membrane proteins, including Type IV Collagen, Laminin, and Entactin [64,65,66,67,68,69].

## 3. Cellular Crosstalk and Paracrine Signaling during Simultaneous Adipogenesis and Angiogenesis 

It is apparent that there is a macrostructural dependence between developing adipocytes and their vasculature, with the vasculature defining the architecture of the lobule and supplying the necessary materials to promote adipogenesis [31,32]. However, there are also paracrine and contact-dependent interactions that promote the development of vascularized adipose tissue. 

It is known that ASCs, which are mesenchymal stem cells not yet committed to the adipocyte lineage, have the ability to differentiate into multiple cell lineages, including adipocytes (white or brown) and endothelial cells [70,71,72,73]. There has been previous histological evidence to suggest that adipocytes and adipose-derived endothelial cells share a common ancestor residing in the vascular niche [34,74,75]. This common precursor was verified and defined by Tang *et al*. through the utilization of a Green Fluorescent Protein (GFP)-Peroxisome Proliferator-activated Receptor Gamma (PPAR- γ) mouse model [76]. This model suggested that there are PPAR-γ positive cells lining the vasculature that also express Smooth Muscle Actin (SMA), PDGF-β, and neuron-glial antigen 2 (NG2), indicating that they are of perivascular origin [76,77]. After isolation, it was shown that these cells took up Bromodeoxyuridine (BrdU), showing their proliferative capacity, while also having immense adipogenic potential [76,78,79]. Together, these findings demonstrate that many adipocytes are derived from progenitors that reside within the mural cells’ compartment, surrounding the vasculature [76,77]. However, there are many regulators that can determine the fate and function of these progenitor cells, including paracrine or mechanical signals.

It should be firstly noted that there is immense crosstalk within the stromal vascular fraction (SVF) that influences both adipogenesis and angiogenesis. One prominent stromal cell, the fibroblast, has demonstrated its importance in adipose tissue development and maintenance. Fibroblast Specific Protein-1 (FSP-1)+ fibroblasts reside adjacent to preadipocytes and regulate PDGF signaling and MMP expression in order to promote adipogenesis [80]. As regulators of matrix construction, these stromal cells also support angiogenesis [81]. ASCs from the SVF promote endothelial colony-forming cell proliferation and differentiation by secreting proangiogenic factors such as VEGF [16,82]. Moreover, those stem cells can then differentiate into pericytes to stabilize the newly formed vessel structure [82]. When in indirect coculture, ASCs release VEGF and Angiopoietin 1 and 2 in order to promote the tubule formation of endothelial progenitor cells which then upregulate their expression of Tunica Interna Endothelial Cell Kinase 2 (TIE2) and VEGFR1. In this study, VEGFR2 was not detected, suggesting a more prominent role in Angiopoeitin-TIE2-mediated angiogenesis in endothelial progenitor cells [83]. 

Endothelial cells within the SVF are also a source of mitogenic and adipogenic components. Conditioned medium from microvascular endothelial cell cultures promoted the proliferation of preadipocytes [84]. It is likely that this media contained heparin-binding Fibroblast Growth Factor (FGF) and Insulin-like Growth Factor-1 (IGF-1), as shown in Figure 2 [85,86]. Furthermore, when endothelial cell-derived ECM components, including Fibronectin, Laminin, and Collagen IV, were applied as soluble entities to differentiating preadipocytes *in vitro*, there was a four-fold increase in triglyceride accumulation, indicating that a strong paracrine signal from endothelial cells is communicated *via* the ECM that they produce [51].

Fukumura *et al*. used an *in vivo* fat pad formation murine model in conjunction with an *in vitro* preadipocyte differentiation assay to monitor simultaneous angiogenesis and adipogenesis as well as unravel the intricacies of the paracrine signaling existing between preadipocytes and endothelial cells [87]. When PPAR-γ is inhibited *in vivo*, 3T3 cells remained undifferentiated, which corresponded to a reduction in vessel infiltration. Furthermore, when VEGFR2 was blocked, there was a reduction in both angiogenesis and 3T3-F442A differentiation. It was found that this was due to the paracrine interaction between endothelial cells and preadipocytes, which is dependent upon endothelial treatments with VEGF, as the treatment with VEGF and a VEGF blocker on preadipocytes did not directly affect differentiation *in vitro* [87]. This study highlights the importance of cell-specific paracrine signaling in simultaneous adipogenesis and angiogenesis. 

In embryonic development and prior to the maturation of white adipocytes, preadipocytes take on a hormonal role and secrete adipokines, including adiponectin [88]. These secretions have been shown to have an imperative paracrine effect in assisting angiogenesis [89]. Leptin, another adipose hormone known to play a regulatory role in satiety, has displayed an important role in regulating fetal growth and organogenesis [90], although its angiogenic implications remain unclear. *In vitro* studies with human umbilical vein endothelial cells (HUVECs) and porcine aortic endothelial cells demonstrated that HUVECs displayed Ob-Ra and Ob-Rb, two isoforms of the Leptin receptor, and responded to Leptin by increasing their proliferative rate and forming capillary-like tubes when in Fibrin gels [91]. This functionality was attributed to the activation of the Mitogen-activated Protein Kinase (MAPK) pathway [91,92]. Other studies have shown the Leptin-mediated upregulation of angiogenic factors FGF-2 and VEGF, further supporting this pro-angiogenic response [93,94]. However, more recently, a study reported that metreleptin, a recombinant leptin analog, does not induce endothelial sprouting in a 3D gel and it also does not affect circulating angiogenic factor levels [89,95]. Additional research shows that adipocyte-derived leptin induces endothelial apoptosis through Ang-2, but angiogenesis occurs through VEGF and FGF-2 (also produced by adipocytes) [93,96,97], demonstrating situationally dependent responses. 

## 4. Mechanical Regulation of Adipogenesis and Angiogenesis 

### 4.1. ECM Regulation 

#### 4.1.1. Stiffness

The *in vivo* microenvironment is constantly modified to preserve the intended function of cells and tissue, necessitating constant cellular mechanosensing (Figure 3). The local microenvironment is a large determinate of the fate of precursor cells, including mesenchymal stem cells, preadipocytes, and endothelial progenitor cells [98]. Investigators mimicked this microenvironment in an effort to better understand the factors that influence adipose tissue development. Young *et al*. sought to recapitulate both the ECM composition and mechanical properties of the native tissue in order to promote adipogenesis in the absence of exogenous small molecules [99]. The bioactive properties of adipose ECM were utilized by functionalizing polyacrylamide gels with decellularized lipoaspirate from women undergoing elective liposuction. ASCs seeded on top of the softest functionalized gels (2 kPa) adopted a rounded morphology and upregulated their adipogenic gene expression, whereas on stiffer (20 and 40 kPa) functionalized gels, stem cells increased their levels of cell spreading [99]. It can be inferred that the stiffness of the microenvironment greatly impacts the lineage specification of stem cells, regardless of the innate adipogenic signaling the matrix holds [57,58]. There has been further evidence to show that these mesenchymal stem cells retain irreversible mechanical memory and experience impaired neurogenesis and adipogenesis when differentiated after switching the culture from a stiff to a soft substrate *in vitro* [100,101,102]. 

The application of external forces on ASCs has also been studied *in vitro*. Under mechanical strain, ASCs increased their proliferation rate, dependent upon the activation of the integrin β1 receptor and the Ras homolog (Rho) family member A (RhoA)/myosin light chain (MLC) pathway [103]. However, the adipogenic potential of these cells was dependent upon the magnitude of the applied force [103,104]. Furthermore, regulating the RhoA pathway has considerable effects on the phenotype and functionality of preadipocytes. Ida *et al*. assessed the effects of Rho-associated coiled-coils containing a protein kinase (ROCK) inhibitor (ROCKi) on three dimensional 3T3-L1 organoids. ROCKi applications increased lipid size, increased Collagen IV and VI, reduced Fibronectin and Collagen I, and produced drastically less stiff organoids [105]. Importantly, the modulation of the Rho pathway can further direct progenitors to either an adipogenic or myogenic cell fate decision, depending on the Rho activity, with lower Rho activity preferentially differentiating towards an adipogenic lineage [106].

Mechanical signals play an important role in regulating adipose vasculature as well. The adipose SVF contains a subset of Fetal Liver Kinase-1 (Flk-1)^+^ endothelial progenitor cells [107,108,109,110]. Importantly, these cells provide paracrine support for angiogenesis while also actively differentiating to participate in vessel formation [107,108,109,110]. It has been demonstrated that cellular specification, towards either endothelial cells or smooth muscle cells, is dependent upon the mechanical environment, with mechanosensing and cell fate decisions being mediated through the Hpo Kinase/Yes-associated protein (Hippo/YAP) pathway [107,108,111,112]. Additionally, VEGF binding, internalization, and signaling amongst endothelial cells is dependent upon matrix stiffness [113]. Endothelial cells on softer substrates internalize more soluble VEGF, the key mediator of angiogenesis [114]. Despite having increased VEGF responsiveness and angiogenic propensity on soft matrices, there is evidence to indicate that endothelial cells and their stromal support cells gradually stiffen the new matrix around the neovessel to promote maturation [115]. However, when initially embedded in a stiff matrix, pathological, sprouting angiogenesis occurs with disrupted endothelial cell–cell junctions [116]. It is apparent that both the adipocyte lineage and endothelial cells depend on a soft environment to allow for proper adipose development.

#### 4.1.2. Matrix Composition

As previously described, and similarly to the gradual occurrence of matrix stiffening, there are progressive alterations in the composition of the extracellular matrix over the course of adipogenesis and angiogenesis. Researchers have sought to examine how these compositions alter the cellular phenotype of adipocytes. Liu *et al*. conducted several studies to establish the effects of collagen remodeling on cellular fate of adipocytes [117]. It was demonstrated that both Collagen I and Gelatin directed 3T3-L1 and C2C12 cells toward a myofibroblast lineage [117]. Importantly, these substrates encouraged YAP nuclear localization and TGF-β1 expression through the activation of reactive oxygen species (ROS) to promote myofibroblast differentiation [117]. Further evaluation of the adipogenic potential of these 3T3-L1 preadipocytes showed that Collagen I inhibits the adipogenesis of 3T3-L1 cells and upregulates primary cilia proteins through YAP inhibition of PPAR-γ and CCAAT enhancer-binding protein alpha (CEBPα) in order to promote cellular migration [118,119,120]. There have also been recent experiments exploring potential extracellular components that regulate thermogenic capacity and the conversion of white adipose tissue towards beige adipose tissue. It has been shown that ASCs induced towards a beige lineage when on top of Collagen 1A1, Collagen 3A1, and Laminin A4 had lower levels of Uncoupling Protein 1 (UCP-1) (a critical component for the regulation of brown/beige adipose tissue) compared to those differentiated on uncoated dishes [121]. In the absence of the Laminin A4 gene, thermogenic markers were upregulated in white adipose tissue, suggesting that the ECM’s composition plays a role in maintaining white adipocytes [121,122,123,124]. 

### 4.2. Tractional Regulation of Cell Fate

It has been stated that changes in the ECM composition and stiffness are able to alter downstream behaviors such as migration, shape, proliferation, and the differentiation of cells. This occurs due to signal transduction through transmembrane proteins, such as integrins and cadherins, which then influence intracellular pathways, cytoskeletal proteins, and transcription factors [125]. As a cell undergoes adipogenesis its matrix composition changes, which in turn affects the surrounding rigidity. As a result, cells remodel their cytoskeleton to define their future fate and function.

#### 4.2.1. Cell Shape 

Mechanical forces simultaneously impact adipogenesis and angiogenesis. White adipocytes are unique in their rounded, spherical shape, which contributes to the overall function of the cell to store lipids. When supported with a simple fibronectin matrix, 3T3-L1 cells will adhere and spread, displaying a low adipogenic potential. However, when forced to stay rounded on the fibronectin, cells will preferentially differentiate towards adipocytes [59]. McBeath *et al*. explained these findings by showing that this shape-dependent differentiation is regulated through the RhoA pathway [59,98]. 

In addition to the substratum regulation of cell spreading, cell crowding and cadherin enactment tend to control cell spreading and shape [126]. Importantly, the cell and ECM density impacts the cell’s spreading capacity, which directly affects the ability for cells, including endothelial cells, to proliferate and differentiate due to the modulation of both Rac and Rho GTPase signaling [126,127,128]. Ingber *et al*. demonstrated that endothelial tube formation is dependent upon several conditions, including a soft substrate with adequate adhesion, cell–cell contacts to allow for retractions from the substrate, and the switch of endothelial cells to a more rounded shape, indicating a cessation in DNA synthetization and a switch towards differentiation [128]. However, the degree of cell rounding was a critical factor as a completely rounded cell would undergo gradual involution and cell death [128]. These endothelial cell-rounding phenomena were found to be dependent upon caldesmon, a component of the actomyosin complex. The inhibition of caldesmon resulted in a reduction in actin stress fibers which encouraged cell rounding that triggered the inhibition of the cell cycle and promoted apoptosis [115]. Additionally, when cell tension was suppressed independently of cell shape, cell growth was also reduced [115]. This indicates that both the cell shape and contractility play imperative roles in defining a cell’s fate.

#### 4.2.2. Tractional Forces

When on top of or encapsulated within a substrate, cells will exert tractional forces onto the substrate. This occurs by forming adhesion complexes and modifying their internal cytoskeletal forces to resist outside forces. It has been shown that the tractional force and cell-spreading areas are proportional to the number of active focal adhesions on the cell, whereby the focal adhesion is the transduction interface between the extracellular environment and the intracellular filaments, commonly *via* an integrin [129].

Preadipocytes are known to be highly motile cells that will migrate to form cellular aggregates within the future lobule throughout the process of development [27]. This process of migration and cellular aggregation is highly dependent on their ability to dynamically remodel the surrounding matrix through cell–matrix interactions in order to then form cell–cell junctions. The preadipocyte mass will respond to both chemical and mechanical signals in order to mature only at the designated adipose lobule [130,131]. As these preadipocytes differentiate, they gradually reduce their tractional forces on the extracellular environment due to the alterations in cell shape and cytoskeletal rearrangement [52,132]. Importantly, adipocytes have the lowest single cell elastic modulus, of 0.61 kPa when compared to osteoblasts and chondrocytes [133]. However, there is evidence to show that murine preadipocytes stiffen with differentiation due to the accumulation of lipids [134,135]. On the contrary, human ASCs undergo a reduction in stiffness over the course of their adipogenic differentiation and cytoskeletal remodeling [52,132,136]. Furthermore, a reduction in ROCK activity promotes adipogenic differentiation and a reduction in cell stiffness in human cells [98,105,136,137]. Additionally, the disruption of actin via cytochalasin D has similar effects to that of Rho Kinase inhibition with Y-27632, whereby adipogenesis is promoted [98]. This indicates that cytoskeletal forces play a large role in the differentiation of adipose cells, with progenitors having disorganized cytoskeletons and lower internal tension being more prone to adipogenesis [98,136,138]. 

Given that the process of angiogenesis and endothelial migration relies heavily on the ability for endothelial cells to overcome external forces by internally balancing the applied forces, they are constantly remodeling their cytoskeleton into filopodia, lamellipodia, or stress fibers [139]. However, this endothelial migration is guided by gradients from proangiogenic chemicals, either soluble or immobilized, or through mechanotaxis, whereby migration is governed by mechanical forces [140,141]. When subendothelial stiffnesses increase, endothelial cells will increase their tractional force and upregulate TGF-β2 [142]. Furthermore, in response to exogenous TGF-β2, endothelial cells will increase their tractional forces [142]. Knizeva *et al*. showed that an increase in ECM density regulates the contractility of endothelial cells (through actin mediated forces) and impairs their angiogenic abilities in 3D [143,144,145].

## 5. Regulation of ECM Remodeling 

### 5.1. Proteolytic Involvement during Adipose Tissue Development

Catabolic ECM remodeling is required to accommodate new vasculature and adipocytes. Thus, proteolytic processes (Table 1), such as the fibrinolytic and matrix metalloproteinase (MMP) systems, are imperative in order to degrade matrix components and activate latent proteinases [42]. 

Firstly, the fibrinolytic system is known to play an important role in angiogenesis as plasmin degrades fibrin and promotes cell migration through the interstitial matrix, as previously mentioned [146,147]. The main inhibitor of plasmin activity is plasminogen activator inhibitor-1 (PAI-1), which mainly targets tissue type-plasminogen activator (t-PA). There has been extensive research into PAI-1 as it is expressed in both human and murine adipose tissue [148,149], although the results have been contradictory. There is literature indicating angiogenic inhibition through VEGFR-2 downregulation and the inhibition of VEGFR-2-vitronectin interactions in HUVECs [150]. This PAI-1 driven inhibition of the vitronectin interaction was similarly observed in human-brain microvascular endothelial cells; however, PAI-1 also stimulated fibronectin dependent migration upon the inhibition of vitronectin interaction [151]. This PAI-1-dependent migratory regulation was similarly observed by Crandall *et al*., who investigated the potential of PAI-1 as a regulator for preadipocyte motility [152]. Interestingly, it was found that preadipocytes and adipocytes synthesize PAI-1, which then binds to the αvβ_3_ integrin receptor on preadipocytes, decreasing their ability to bind to vitronectin and reducing their ability to migrate [152,153]. Further, it has been shown that the overexpression of PAI-1 in 3T3-L1 cells inhibits their adipogenic differentiation, as indicated by a reduction in PPARγ, C/EBPα, and aP2, while also inhibiting plasmin and increasing Collagen I [154]. This indicates a complex, potentially haptotactic role of PAI-1 in adipose development that should be explored further.

Matrix Metalloproteinases (MMPs) are another group of proteases that are able to degrade a range of matrix proteins, activate other MMPs, process bioactive molecules, bind to surface receptors, and regulate gene expression [155]. These MMPs are highly regulated by tissue inhibitors of MMPs (TIMPs), and the overall degradation of the tissue is determined by the balance between MMPs and TIMPs. These metalloproteinases are critical for adipogenesis and angiogenesis to occur during adipose development [156]. The most well-studied MMPs in adipose tissue are MMP2, MMP9, and MT1-MMP. Both MMP2 and MT1-MMP have exhibited immense adipogenic potential and are promotors of white adipose-tissue development [42,157,158]. MMP2 is secreted by adipocytes and promotes their subsequent migration and organization into three dimensional clusters within a completely remodeled matrix [157]. Chun *et al*. showed that MT1-MMP’s adipogenic potential lies in its ability to degrade collagen to coordinately alter matrix compliance, cellular tension, and subsequent cell shape. These matrix degradation products were further shown to increase adipocyte hypertrophy [98,159].
biomedicines-10-02227-t001_Table 1Table 1Proteolytic Enzymes, their regulators, and downstream effects.Proteolytic SystemEnzymesActivatorsInhibitorsActivityCitationFibrinolyticPlasminogen-inactive▪Tissue-type plasminogen activator (t-PA) (Predominantly in blood circulation)▪Urokinase-type plasminogen activator (u-PA) (Activates cell bound plasminogen)▪α2 antiplasmin▪Plasminogen activator inhibitors (PAI-1/2)▪Degrades Fibrin into soluble degradation products▪Activates some MMPs[147,152,156]Plasmin-ActiveCollagenasesMMP1(Collagenase 1)▪VEGF induces MMP1 expression▪TIMPs▪Digest Fibrillar Collagen▪ASC-derived MMP1 promotes vascular sprouting[160,161,162]MMP13(Collagenase 3)▪MT1-MMP activates proMMP13▪Digest Fibrillar Collagen▪Enhances over the course of adipose differentiationMT1-MMP▪Regulated by proprotein convertase▪Inhibited by TIMP2▪Degrade Type 1 Collagen to promote three-dimensional adipocyte lipid accumulation through matrix rigidity regulation▪Promotes tubulogenic activity of endothelial cells[158,159,163]GelatinasesMMP2(Gelatinase A)▪Direct activation of proMMP2 (progelatinase A) is achieved in a plasmin independent mechanism▪Activated through TIMP2 and homo-dimerization of MT1-MMP▪During angiogenesis-serine proteinase plasmin activates▪Ro 28-2653▪Tolylsam▪TIMP-1 (B)▪TIMP-2 (A)-without MT1-MMP▪Fragments Basement Membrane, Collagen, and Gelatin promoting endothelial proliferation, chemotaxis, and angiogenesis▪Mediates adipocyte migration and clustering to form lobular architecture[42,43,158,164,165,166,167]MMP9(Gelatinase B)▪Activation of proMMP9 (progelatinase B) is achieved by plasmin▪TIMP’s▪Release of fibrinogen and fibronectin which lay the provisional foundation for the migration of additional endothelial cells▪Promotes adipocyte differentiation[40,41,42,43,44,89]

### 5.2. Adipokine Regulation of ECM Remodeling 

As aforementioned, adipokines are critical hormones released primarily from adipose tissue that play a role in fetal development, as gestational levels of adipokines predict future adipose growth patterns [168]. These adipokines, notably Adiponectin and Leptin, have a diverse set of targets and responses. One cellular response is extracellular remodeling for the purpose of adipogenesis or angiogenesis in tissue development and expansion. Importantly, it has been shown that Leptin treatment increases the expression of proteinase, MT1-MMP, which then translocates to the plasma membrane to mediate matrix proteolysis and the activation of gelatinases, MMP2, and MMP9 [169,170]. The activity of MMP2 was then found to be critical for proper adipogenesis to be undertaken [169,171]. Similar effects were found with the treatment of Leptin on myofibroblasts, whereby Leptin treatment promoted MMP2 concurrently with MT1-MMP and promoted intracellular accumulation of pro-collagen-1, a precursor to Collagen I [172]. 

However, it has been shown that Adiponectin has antagonizing effects on endothelial angiogenic activity. Adiponectin agonist, AdipoRon, promotes endothelial migration and tube formation, while also upregulating CXCL1, VEGF-A, MMP2, and MMP9 in HUVECs [89]. This angiogenic activity is due to the combination of chemotactic signaling alongside the facilitated matrix organization [89,170]. In cardiac fibroblasts, adiponectin promotes cellular migration through APPL1-AMPK signaling, inducing collagen and elastin remodeling but maintaining basal collagen secretion [170]. Similarly, in human microvascular endothelial cells, globular adiponectin binds to the AdipoR1 receptor and promotes cellular proliferation and migration through the AMPK-Akt pathways [173]. While MMP2 and MMP9 activation was mediated by globular adiponectin binding to either AdipoR1 or AdipoR2 receptors, full-length adiponectin promotes microvascular endothelial cell proliferation through the AdipoR2 receptor [173]. It is evident that developing adipose tissue regulates, through adipokine release, the progression of adipogenesis and angiogenesis through extracellular matrix-mediated communication. 

## 6. Conclusions

Despite being accepted as a critical connective tissue, adipose tissue has only recently been recognized for its paracrine and endocrine contributions. In the past several decades, these functionalities have been investigated specifically during adipose development, but there has yet to be a comprehensive explanation of the appropriate cellular and matrix interactions that exist during adipose development.

Adipose tissue development is a coordinated process that is highly dependent on matrix structural components, paracrine signaling, mechanical regulatory signals, and proteolytic degradation that is undertaken during simultaneous adipogenesis and angiogenesis. Here, the spatiotemporal relationship between adipogenesis and angiogenesis has been highlighted. Early reports of human buccal adipose-tissue development demonstrate that during the second trimester of human gestation, the foundation for white adipose tissue is laid with newly synthesized vascular structures providing both a structural and functional route for adipogenesis [27,28,174]. Additionally, the developing adipose tissue provides feedback to the sprouting vasculature in the form of paracrine signals. These signals, including adipokines, such as Leptin and Adiponectin, regulate endothelial cellular activities such as proliferation in addition to encouraging proper proteolytic activity to support the appropriate environment for developing adipocytes [89]. Several factors of this microenvironmental regulation are illustrated. Importantly, extracellular matrix stiffness, rigidity, and composition play an important role in both endothelial and adipocyte cell fate [19]. Through mechanotransduction, both adipocytes and endothelial cells sense and respond to external forces by adjusting their cytoskeletal tension and cellular traction. Through their applied traction, cells are able to migrate, which is critical for preadipocytes and endothelial cells during adipose lobule formation [125]. It is crucial to gain a better understanding of these complex relationships that exist between adipose tissue and its vasculature during prenatal and postnatal adipose tissue development. This information has the potential to provide insights into the cellular and biomechanical activities of adipose expansion for applications regarding soft-tissue reconstruction or the development of pathological obesity. 

## Figures and Tables

**Figure 1 biomedicines-10-02227-f001:**
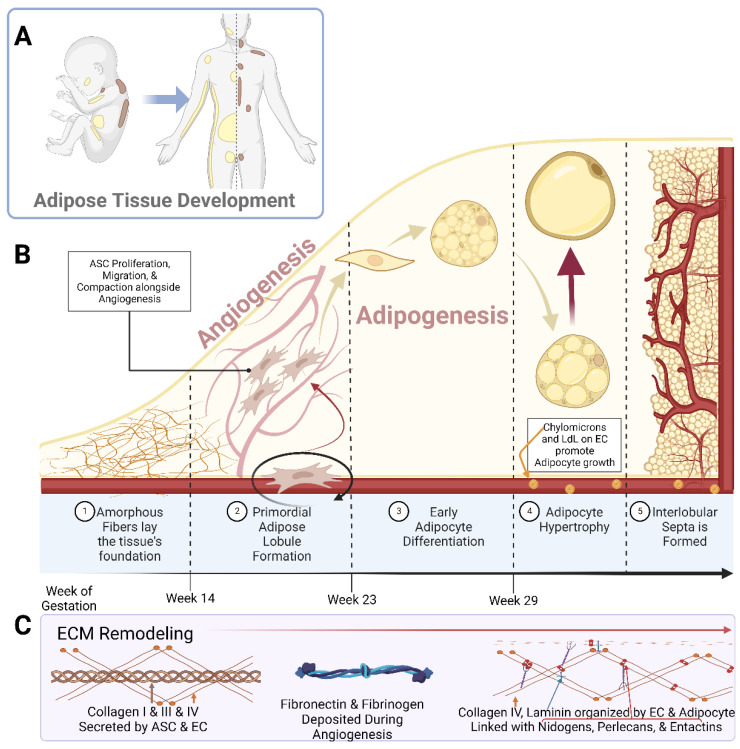
Schematic representations of (**A**) the localization of white adipose tissue (yellow) and brown adipose tissue (brown) in a developing neonate and adult human; (**B**) the breakdown of adipose tissue development into the 5 specified stages; (**C**) the prominent Extracellular Matrix (ECM) components over the course of adipose tissue development. **Abbreviations:** adipose stem cell (ASC), low-density lipoprotein (LdL), and endothelial cell (EC). Created with Biorender.com.

**Figure 2 biomedicines-10-02227-f002:**
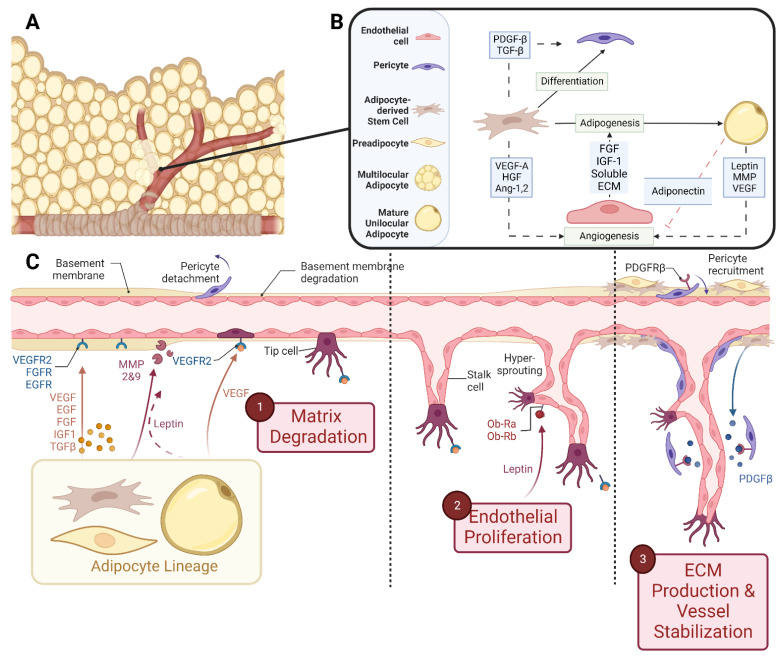
Schematic representations of (**A**) vascularized adipose tissue and (**B**) paracrine interactions between both the vascular and adipose lineages. (**C**) Stepwise breakdown of cellular and paracrine activity during angiogenesis. **Abbreviations:** Platelet-derived growth factor-β (PDGF-β), transforming growth factor-β (TGF-β), vascular endothelial growth factor-A (VEGF-A), hepatocyte growth factor (HGF), angiopoietin-1,2 (Ang-1,2), fibroblast growth factor (FGF), insulin-like growth factor-1 (IGF-1), extracellular matrix (ECM), and matrix metalloproteinase (MMP). Created with Biorender.com.

**Figure 3 biomedicines-10-02227-f003:**
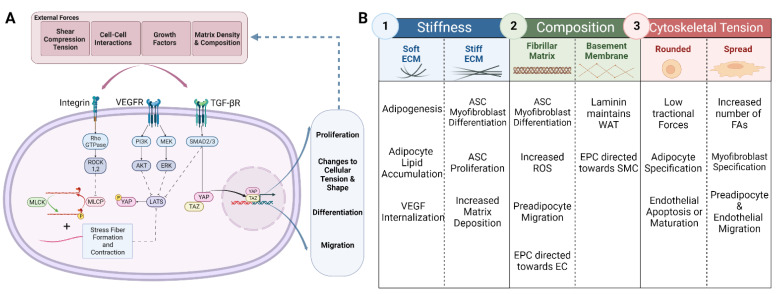
Schematic representations of (**A**) major mechanotransduction pathways and their downstream effects (**B**). Mechanoresponses of adipose derived stem cells (ASC), adipocytes, endothelial progenitor cells (EPC), and endothelial cells (EC) in situations of (1) altered matrix stiffness, (2) altered matrix composition, and (3) altered internal forces. **Abbreviations:** Vascular endothelial growth factor (VEGF) receptor (VEGFR), transforming growth factor-β receptor (TGF-βR), Extracellular matrix (ECM), Ras homologous GTPase (Rho GTPase), Rho-associated protein kinase (ROCK), myosin light chain phosphatase (MLCP), myosin light chain kinase (MLCK), phosphatidylinositol 3- kinase (PI3K), protein kinase B (Akt), mitogen-activated protein kinase kinase (MEK), extracellular-signal-regulated kinase (ERK), large tumor suppressor (LATS), small mothers against decapentaplegic (SMAD), yes-associated protein (YAP), transcriptional coactivator with PDZ-binding motif (TAZ), reactive oxygen species (ROS), white adipose tissue (WAT), smooth muscle cell (SMC), and focal adhesion (FA). Created with Biorender.com.

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
