# Peer review of "Adipose Tissue Development Relies on Coordinated Extracellular Matrix Remodeling, Angiogenesis, and Adipogenesis"

_biomedicines, 2022, doi:10.3390/biomedicines10092227_

Round 1

Reviewer 1 Report

The current manuscript authored by Elizabeth K. Johnston and Rosalyn D. Abbott described “Adipose Tissue Development Relies on Coordinated Extracellular Matrix Remodeling, Angiogenesis, and Adipogenesis”.

The present review explores the process of adipose tissue development. This is an interesting issue and excellent review. However, there are several concerns that should be addressed.

1.      Please change “peroxisome proliferator-activated receptor gamma …” to “ Peroxisome Proliferator-activated Receptor Gamma  ...” in page 5, line 171.

2.      Please change “platelet-derived growth factor-β (PDGF)-β” to “ PDGF-β.” in page 5, line 172.

3.      The Table 1 with horizontal may be easy reading to readers.

4.      Please provide the page No. of Reference 1, 4, 17, 70, and 83.

5.     Please complete the page No. of Reference 32, 33, 34, 42, 43, 44, 53, 67, 68, 71, 72, 73, 75, 79, 85, 91, 92, 94, 97, 98, 103, 127, 128, 129, 148, 151, 157, 159, 166, 167, 171, and 174. (For example, the page No. of Reference 32 was shown as 315-326. )  

Author Response

Thank you for your attention to details! We have made all of the suggested changes.

Reviewer 2 Report

In this review article, authors summarized recent researches investigating the adipose tissue development and ECM remodeling / angiogenesis / adipogenesis. This review is well-written and covers sufficient current information about this topic. This review also provides the important insight into potential future clinical indications.

I believe this review article should be accepted.

Author Response

Thank you for taking the time to read it. We greatly appreciate your feedback!

Reviewer 3 Report

This is a well written review on adipose tissue and its development and remodeling. The review begins with adipogenesis, cellular cross-talk and paracrine signaling. These processes are clearly defined and beutifully illustrated in the figures. The review also discusses the extra cellular matrix and its role in adipose tissue remodeling. The review is comprehensive and is an important summary of the current status of the field.

Author Response

(The authors gave the same response as above.)
